# Olfactory markers for depression: Differences between bipolar and unipolar patients

**François Kazour**[1,2,3]*, **Sami Richa**[3], **Chantale Abi Char**[2], **Alexandre Surget**[1], **Wissam Elhage**[1,4☉], **Boriana Atanasova**[1☉]

**1** UMR 1253, Inserm, iBrain, Université de Tours, Tours, France, **2** Psychiatric Hospital of the Cross, Jal Eddib, Lebanon, **3** Department of Psychiatry, Faculty of Medicine, Saint-Joseph University, Beirut, Lebanon, **4** CHRU de Tours, Clinique Psychiatrique Universitaire, Tours, France

☉ These authors contributed equally to this work.
* francoiskazour@hotmail.com

**Data Availability Statement:** All relevant data are within the paper and its Supporting Information files. Any additional information on the database can be requested at any time from the authors of this study.

## Abstract

### Objectives

The aim of the study was to compare olfactory functions between unipolar and bipolar patients according to the thymic states (depressed, euthymic) and determine specific olfactory variations between these different states.

### Methods

We recruited 176 participants in 5 groups: depressed bipolar (DB), euthymic bipolar (EB), depressed unipolar (DU), euthymic unipolar (EU), and controls (HC). They were assessed using the Sniffin' sticks threshold and identification tests. Odors' pleasantness, intensity, familiarity and emotion were assessed. Clinical evaluation explored dimensions of depression, mania, anxiety, and anhedonia.

### Results

Smell identification was lower in DU compared to EU patients and controls. Pleasant odors received lower hedonic rating in DU and DB patients compared to EU and EB patients respectively. Negative correlation was found in EB patients between hedonic rating and social anhedonia. In EU patients hedonic rating was negatively correlated with anxiety-state, and anhedonia.

### Conclusions

Odor identification of pleasant odors is altered in both depressive states. Only unipolar patients would recover a regular identification level in symptomatic remission, while bipolar subjects would keep their deficits. Hedonic rating is lower in bipolar depressed patients compared to unipolar ones, and these deficits improve after remission. Hedonic rating of pleasant odors may distinguish bipolar depression from unipolar depression during periods of decompensation and phases of remission. Olfactory assessment may be useful to screen unipolar and bipolar depression, leading to possible future sensory markers in mood disorders.

**Funding:** The authors received no specific funding for this work.

**Competing interests:** The authors have declared that no competing interests exist.

## Introduction

Depressive episodes can be a presentation of either Major Depressive Disorder (MDD–Unipolar Depression) or Bipolar Disorder (BPD) [1]. BPD differs from MDD in the occurrence of (hypo)manic episodes. Although some symptomatic differences may exist between unipolar and bipolar depression, it is still difficult to decide upon the appropriate clinical diagnosis [2]. Mitchell et al. (2008) recommends a probabilistic approach to differentiate clinically unipolar from bipolar depressive episodes. This is why it is difficult to determine if "typical, non-psychotic" depression is of unipolar or bipolar nature in the absence of history of (hypo)mania. Therefore, a more reliable clinical measure differentiating between unipolar and bipolar depression is needed to have a more accurate diagnosis and hence a better treatment plan and prognosis.

Several sensory variables have been identified as potential markers of unipolar depression. Deficits in visual perception (retinal contrast gain, spatial suppression, visual attention) [3–5] and alterations in auditory measures (auditory evoked potentials, auditory processing) [6–9] have been found to be potential markers of depression. Taste perception can also be altered in MDD [10]. Taste sensitivity to bitter compounds has been proposed as a potential marker for depression and anhedonia [11]. As for olfaction, studies showed that odor sensitivity and hedonicity can be altered in depression [12–13]. However, most studies on sensory markers did not differentiate between unipolar and bipolar depressive episodes.

Concerning MDD, several studies demonstrated a reduced olfactory sensitivity in depressed patients [14–18]. However, this impairment in olfactory acuity recovered with the symptomatic remission of depression [14, 16]. According to Croy and Hummel (2017), olfactory function impairment in depression is a result of a diminished olfactory attention and reduction in olfactory receptor turnover rates. The authors also considered that reduction in the olfactory bulb volume can constitute a marker of vulnerability for depression [19].

As for odor identification, an indicator of central olfactory processing, most studies showed the absence of alteration in depressive episodes [15, 20, 21]. However, this finding is controversial since other studies showed that depressed patients can exhibit lower levels of odor identification [22]. As for perceived odor intensity, no significant difference was found in depressed patients compared to controls [14, 15]. However, hedonic rating may be affected by depression since depressed patients over-evaluated the pleasantness of positive odors, suggesting a functional bias in brain processing of pleasantness in depressive states [14, 15]. Atanasova et al. (2010) showed that hedonic perception of unpleasant odors is also impaired, with depressed patients perceiving the unpleasant odorant as more unpleasant than controls (olfactory negative alliesthesia). Depressed patients were also unable to discriminate between different concentrations of pleasant odor, thus having an "olfactory anhedonia" [23]. Naudin et al. (2012), suggested that "olfactory anhedonia" (expressed by decrease of hedonic score) may be a state marker of depressive episodes, while "olfactory negative alliesthesia" may be a potential trait marker of depression persisting after clinical remission [24].

Among all studies evaluating olfactory function in depression, very few compared unipolar and bipolar depressive episodes [10]. Lövdahl et al. (2014), showed that 14% of patients with BPD type 2 disorder, and 17.5% of patients with depression within the bipolar spectrum have an impaired sense of smell, compared to 0% of controls [25]. Decline in olfactory sensitivity may constitute a differentiation marker between these two types of episodes since it has been found in unipolar depression and not in bipolar one [26]. Lahera et al. (2016), showed that euthymic bipolar patients have an impairment in olfactory identification compared to healthy controls [27]. Swiecicki et al. (2009) compared patients with unipolar *versus* bipolar depression and found that unipolar patients rate less olfactory stimuli as pleasant compared to patients

with BPD. However, no difference was found between groups in olfactory threshold, olfactory identification, and the number of odors rated as unpleasant or neutral [21]. According to Parker (2104), bipolar subjects experience suprasensory changes of smell and taste during their manic/hypomanic states. According to the author, these changes are more frequent in bipolar II patients compared to those with bipolar disorder type I. These sensory changes attenuated or disappeared during depressive and euthymic phases [28, 29]. Olfactory acuity in BPD is related with psychosocial and cognitive performances. Indeed, BPD patients with lower levels of fear and avoidance may exhibit a better odor sensitivity [12]. Considering the differences in the results of the studies mentioned above, and the scarcity of studies comparing olfactory performance between unipolar and bipolar depression, in the present study we assessed olfactory function in patients with unipolar or bipolar depression, in symptomatic or euthymic states.

The primary objective of this study was to find differences in olfactory perception (olfactory threshold and identification) between patients in unipolar and bipolar depressive states and controls, and to find if these differences persist after remission of depressive episodes. The secondary objectives were the following: to determine bipolar and unipolar patients' judgments of different aspects of olfaction (pleasantness, intensity, familiarity and emotional aspect) and to study the correlations between these olfactory judgments and the clinical patients' state (severity of depression, anhedonia, anxiety).

The primary hypothesis of this study was that olfactory threshold and identification capacity would be altered in depressed individuals compared to healthy controls. This alteration would depend on the type of depression. As for the secondary hypotheses of this study, they were the following:

- Depressed patients compared to healthy may show deficits in assessing the hedonic value of olfactory stimuli.

- The judgments of smells' familiarity, intensity and emotional aspect may be lower in depressed patients compared to controls.

- The olfactory deficits seen in depressed patients may differ between unipolar and bipolar subjects and between subjects in symptomatic phases and in remission, thus constituting potential differentiation markers of bipolarity.

## Material and methods

### Participants

Patients were recruited in the inpatient and outpatient psychiatric units of two hospital settings (Psychiatric Hospital of the Cross, and Hôtel-Dieu de France, Lebanon). Patients were divided into 4 clinical groups: Depressed Unipolar (DU), Depressed Bipolar (DB), Euthymic Unipolar (EU) and Euthymic Bipolar (EB). The healthy controls (HC) were recruited among individuals with no history of any mood or psychotic disorder or any psychiatric treatment. Groups were matched based on key demographic characteristics of participants such as age, sex and smoking status (see part "Results", "Demographic and clinical characteristics").

Inclusion criteria for patients were the following: age between 18 and 64 years, current or past diagnosis of a depressive episode, absence of smell impairment related to any brain or nasal surgery or lesion, absence of current pregnancy, absence of current or past substance use disorders, in remission for more than 3 months for patients included in the euthymic groups. Concerning the substance use aspect, the smokers were included in the study, because half of

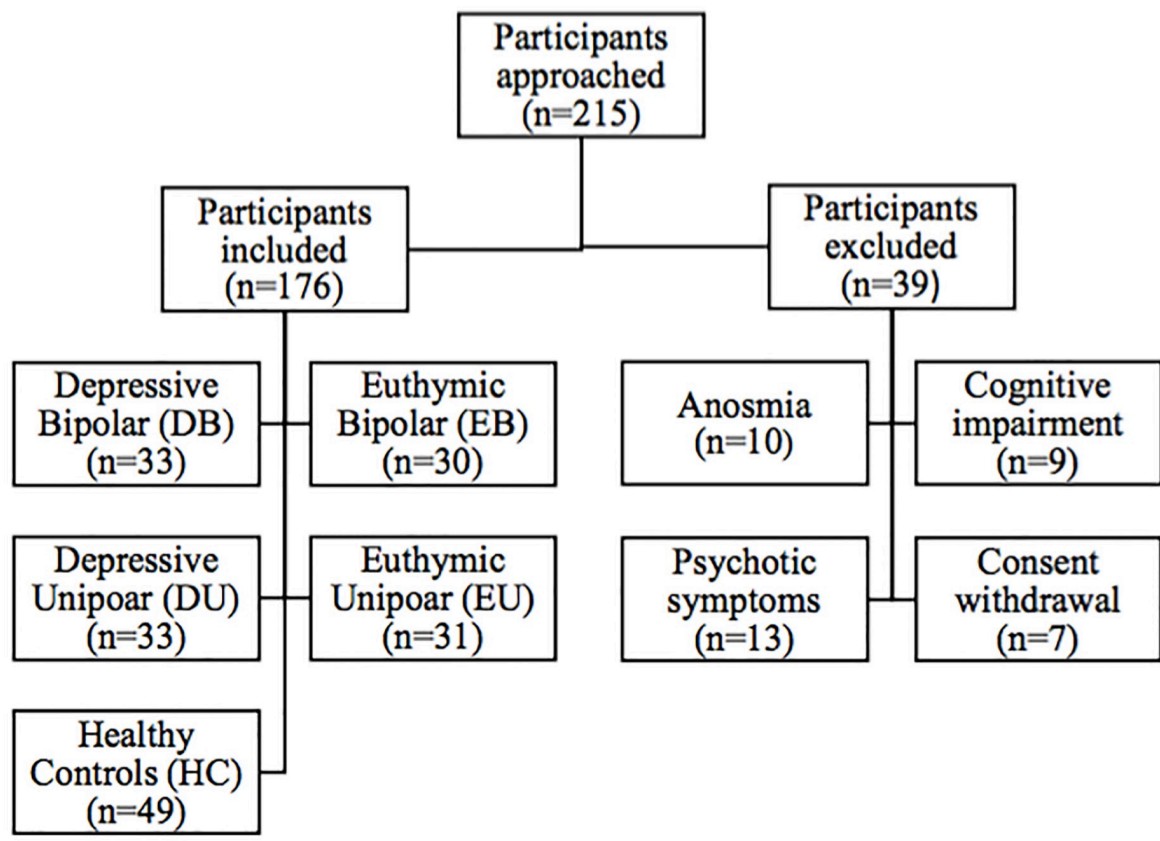

**Fig 1. Distribution of groups of participants.**

the patients in each group presented tobacco abuse. However, knowing that this parameter may be altered olfactory perception; all groups were matched concerning this aspect. Euthymic subjects had to be symptom free for at least 3 months prior to evaluation, with a MADRS (Montgomery-Åsberg Depression Rating Scale) score lower than 9 on assessment [30, 31]. Exclusion criteria for all participants (patients and controls) were the following: presence of any psychotic symptom, presence of (hypo)manic or mixed episodes, severe cognitive impairment, treatment with medication affecting olfaction, inability to undergo the assessment and anosmia (inability to smell the highest odor concentration on the olfactory threshold test used in the study). Some medications may affect olfaction or induce smell complaints. The most used medications with such effects are the following: Amoxicillin, Azithromycin, Ciprofloxacin, Fluticazone, Prednisone, Amlodipine, Diltiazem, Enalapril, Atorvastatin, Lovastatin, Pravastatin, Levothyroxine [32, 33]. Subjects included in this study took only psychotropic medications for their affective disorders, and none of the drugs listed above.

Over 18 months, 215 participants were approached (Fig 1), and 176 participants were included in five groups: DU (n = 33), DB (n = 33), EU (n = 31), EB (n = 30), and HC (n = 49). Thirty-nine participants were excluded for the following reasons: anosmia (n = 10), cognitive impairment (n = 9), presence of psychotic symptoms (n = 13), and consent withdrawal (n = 7).

The study was approved by the local ethical committee board (Faculty of Medicine, Saint-Joseph University, Beirut, Lebanon) and conducted in accordance with Good Clinical Practice procedures and the current revision of the Declaration of Helsinki. All participants signed an

informed consent. The two evaluators in this study were a clinical psychiatrist and clinical psychologist both trained to use the scales and tests needed for this study.

## Clinical assessment

All participants had a 90–120 minutes' interview to assess their clinical status and their olfactory function. Interviewers obtained information concerning social and demographic status (age, marital status, educational level, working status), present and past medical history, current and past treatments, number of depressive episodes, number of manic and hypomanic episodes, total duration of depressive episodes, number of hospital admissions, age of onset of mood disorder and smoking status.

Clinical assessment included the following tools: The Mini International Neuropsychiatric Interview (MINI 5.0.0) (Arabic validation) [34, 35] was used for the diagnosis of current and past psychiatric disorders; the Montgomery-Åsberg Depression Rating Scale (MADRS) (Arabic validation) [30, 36] was used to assess the severity of depressive symptoms; the Young Mania Rating Scale (YMRS) [37] was used to confirm the absence of any manic, hypomanic or mixed episodes; the State-Trait Anxiety Inventory (STAI) (Arabic validation) [38, 39] was used to evaluate the intensity of anxiety symptoms; and the Chapman physical and social anhedonia questionnaire [40, 41] was used to evaluate clinical anhedonia.

## Olfactory assessment

Olfactory tests evaluated patients' olfactory threshold, olfactory identification capacity and their rating of odors hedonic aspect, familiarity, intensity, and emotional impact.

The smell threshold is the minimum concentration at which an odor is perceived. It corresponds to the measure of the lowest concentration of a particular olfactory stimulus required to activate the olfactory receptors. The Sniffin' sticks threshold test (Burghardt®) [42] was used to determine the threshold by the "staircase procedure". The test consists of the successive presentation to the subject of a triplet of 3 "odor sticks". One stick contains a given concentration of phenyl-ethanol (rose-like odor) dissolved in propylene glycol, whereas the others contain the propylene glycol alone. The test contains 16 triplets of sticks with increasing concentrations of phenyl-ethanol. Starting with the lowest concentration, the subject was asked to report which of the 3 sticks contains the odor. If an incorrect response occurred on this trial, the higher concentration is presented. When two consecutive correct trials occurred at a given concentration, the subsequent stimulus was presented one concentration lower to determine if it can be correctly perceived. If one of 2 trials was missed, the examiner switched back to the higher concentration. After a series of 7 switches between concentrations, the geometric mean of the last four staircase reversal points was used as the threshold estimate. High scores of the threshold test reflected high sensitivity to odors.

Odor identification was tested using the Sniffin' sticks identification test–Screening 12 Test (Burghardt®, Wedel, Germany) [43]. A series of 12 "odor sticks" (banana, cinnamon, clove, coffee, fish, leather, lemon, liquorice, orange, peppermint, pineapple, rose) was presented to the subject. Each time, the subject had to identify the odorant from a list of four descriptors (multiple choice paradigm). The score of 1 or 0 was attributed when the odor was correctly and incorrectly identified respectively. The maximum identification score was 12. Then the subject had to evaluate the pleasantness (hedonic aspect), the familiarity level, the intensity and the emotional rating of the perceived odors on a 10 cm linear scale labeled at each end (highly unpleasant/highly pleasant; unfamiliar odor/very familiar odor; low intensity/very intense; negative emotion/positive emotion). The resulting response was expressed with a score ranging from 0 to 10.

Moreover, it has been demonstrated that anhedonia (one of the two cardinal symptoms of depression) can be detected in some psychiatric disorders through the use of odorants with opposite hedonic valence [13, 23]. Consequently, in the present study, the 12 odors of the used standardized Sniffin' sticks—Screening 12 identification test were divided between pleasant (hedonic score more than 5) and unpleasant (hedonic score less than 5) according to controls ratings (Positive odors (POS): banana, cinnamon, coffee, lemon, liquorice, orange, peppermint, pineapple and rose) and unpleasant (Negative odors (NEG): clove, fish and leather). This permitted also to increase the power of the results and to test the influence of odor's hedonic aspect on the olfactory perception.

## Statistical analysis

First of all, the sample size calculations were performed based on our preliminary data for olfactory identification and in order to have a number of participants for a statistical power of 0.8, allowing the detection of an effect size $eta^2 = 0.06$ (i.e. d = 0.5) at a significance threshold of 0.05. Based on our sample size (n $\geq$ 30 per group), conditions for applying the central limit theorem was met, allowing us using one-way ANOVA even when normality may not be ensured for all samples.

The Chi-square test was used to compare proportions of qualitative variables of the different groups of subjects (sex, smoking status and number of correct identification responses per odors or group of odors: POS and NEG). The Marascuilo procedure was used to carry out comparison of all possible pairs of proportions between groups for number of correct identification responses per odors or group of odors: POS and NEG. The quantitative variable (age, educational level, total identification score, threshold score, age of onset, number of depressive episodes, number of hospital admissions, MADRS score, STAI-state score, STAI-trait score, YMRS score, social anhedonia score and physical anhedonia score) of the five groups (or four groups, when only the patients' groups were compared) were computed separately with an analysis of variance (ANOVA) with 1 factor: group. As significant effect of group was found, a two-by-two comparison between groups was carried out using Tukey post hoc test.

For each odor's characteristic (pleasantness, familiarity, intensity and emotion), analysis of variance with 2 factors: stimulus (POS and NEG odors) and group (5 groups of subjects: DU, DB, EU, EB and HC) and their interaction (group×stimulus) was carried out. When significant effects of stimulus, group or group×stimulus interaction were found, a two-by-two comparison between groups for each stimulus was carried out using Tukey post hoc test.

The Pearson correlation coefficient was used to study the relationship between the clinical subjects' state and their olfactory performances. The Pearson coefficient was calculated for the 4 patients' groups and the significant results obtained in the different tests and scales.

Finally, receiver operating characteristic (ROC) curve analyses were carried out based on the group division (patient groups versus healthy controls). The accuracy (area under the curve, AUC) to which the olfactory tests can predict whether a person belongs to the healthy controls or to any other group were reported. AUC value can vary between 0 and 1. An excellent predictor test would display an AUC near to 1, which means it provides a strong measure of separability between groups (patients and healthy controls), i.e. high specificity and high sensitivity. When AUC is near 0.5, it means the test has no class separation capacity whatsoever and is uninformative. A poor predictor test has AUC near to the 0 which means it has worst measure of separability. A z-test was used to compare each AUC to 0.5 allowing checking if the diagnostic test is more powerful than just a random rule. The ROC curve was

generated and the AUC was calculated for all olfactory tests: threshold, identification, pleasantness (POS and NEG odors), familiarity, intensity and emotional rating. Concerning pleasantness judgment, the analysis was carried out for POS and NEG odors, because the two-way ANOVA revealed a significant "group x stimulus" interaction for this variable. For each olfactory test, the AUC of the four patients' groups was compared using Student test for independent samples. Pairwise comparisons were Bonferroni-corrected for multiple comparisons between the four groups.

All statistical analyses were performed at 95% confidence interval (alpha = 5%). They were conducted using XLstat-Pro software. The effect sizes are reported as η2 or Cohen's d, for ANOVA and Student tests respectively. The statistics of the Tukey post hoc tests (p-values and effect sizes), the ROC curves and the optimal cutoff values for each olfactory test and groups of subjects are reported in S1–S13 Tables.

## Results

### Demographic and clinical characteristics

The 5 groups of participants were matched according to age ($F_{(4,171)} = 0.2$, p = 0.95), sex ($\chi^2 = 0.1$; df = 4; p = 1) and smoking status ($\chi^2 = 0.8$; df = 4; p = 0.94) since these variables may be factors of confusion affecting olfactory perception (Table 1). No significant difference between the groups was found also for the educational level ($F_{(4,171)} = 2.2$, p = 0.07).

The mean number of (hypo)manic episodes was of 3.8 (5.1) and 3.2 (4.0) in the depressed (DB) and euthymic bipolar (EB) groups respectively while none of the controls (HC) or unipolar participants, either depressed (DU) or euthymic (EU), experienced any (hypo)manic episode. Concerning patients, a significant group effect was highlighted for the number of depressive episodes ($F_{(3,123)} = 7.1$, p<0.001, $\eta^2 = 0.15$) and for the number of hospital admissions ($F_{(3,123)} = 8.9$, p<0.001, $\eta^2 = 0.18$). The mean number of depressive episodes was of 8.8 (10.9) for DB, significantly higher than other groups (4.6 (4.2) for EB, 3.5 (3.7) for DU and 2 (1.5) for EU) (Table 1). There was no difference between patients' groups concerning age of onset ($F_{(3,123)} = 1$, p = 0.4, $\eta^2 = 0.02$). As expected, a significant group effect was found for all clinical and psychometric parameters (MADRS: $F_{(4,171)} = 507$, p<0.001, $\eta^2 = 0.23$; YMRS: $F_{(4,171)} = 2.9$, p = 0.02, $\eta^2 = 0.06$; Physical Anhedonia: $F_{(4,171)} = 12$, p<0.001, $\eta^2 = 0.22$; Social Anhedonia: $F_{(4,171)} = 12$, p<0.001, $\eta^2 = 0.28$; STAI-state: $F_{(4,171)} = 60$, p<0.001, $\eta^2 = 0.58$; STAI-trait: $F_{(4,171)} = 29$, p<0.001, $\eta^2 = 0.40$). The Tukey post hoc tests showed that the scores on MADRS, anhedonia and STAI scales were significantly higher in depression groups compared to both euthymic groups and controls (Table 1). Statistics of the post hoc tests (p-values and effect sizes) concerning demographic and clinical characteristics of patients are reported in the S1–S13 Tables.

### Olfactory threshold

Concerning the olfactory threshold, a significant group effect was found ($F_{(4,171)} = 2.6$, p = 0.036, $\eta^2 = 0.06$). The two-by-two comparisons between groups showed that DU patients have a significantly lower sensitivity to odor compared to controls. Olfactory threshold scores were not significantly different with DB, EB and EU groups (Fig 2). Overall, we observed a tendency showing a progressive increase in odor sensitivity between depressed (unipolar and bipolar) states, euthymic (unipolar and bipolar) states and healthy control state respectively. Statistics of the post hoc tests (p-values and effect sizes) concerning olfactory threshold are reported in the S1–S13 Tables.

**Table 1. Demographic and clinical characteristics of participants.**

| Patients' Groups | DB (n = 33) | EB (n = 30) | DU (n = 33) | EU (n = 31) | HC (n = 49) |
|---|---|---|---|---|---|
| Mean age, SD | 36.6 (10.3) | 36.2 (13.4) | 36.8 (9.9) | 34.9 (14.0) | 35 (12.1) |
| Female/male, ratio | 25/8 | 23/7 | 25/8 | 24/7 | 38/11 |
| Smokers/non-smokers, ratio | 16/17 | 15/15 | 17/16 | 14/17 | 21/28 |
| Educational level, mean (SD) * | 2.1 (0.7) | 3.5 (5.2) | 2.1 (0.7) | 2.7 (0.6) | 3.0 (0.2) |
| Marital status, % | | | | | |
| • Single | 54.5 | 60 | 45.5 | 74.2 | 63.3 |
| • Married | 36.4 | 23.3 | 45.5 | 22.6 | 36.7 |
| • Divorced | 9.1 | 10 | 3 | 3.2 | 0 |
| • Widowed | 0 | 6.7 | 6 | 0 | 0 |
| Age of onset, mean (SD) | 24.5 (8.7) | 25.1 (11.3) | 28.3 (9.6) | 24.9 (10.9) | - |
| Depressive episodes, n (SD) | 8.8 (10.9) [B] | 4.6 (4.2) [A] | 3.5 (3.7) [A] | 2 (1.5) [A] | 0 |
| (Hypo-) Manic episodes, n (SD) | 3.8 (5.1) | 3.2 (4.0) | 0 | 0 | 0 |
| Hospital admissions, n (SD) | 4.5 (6.1) [B] | 1.5 (2.7) [A] | 1.7 (1.9) [A] | 0.1 (0.4) [A] | 0 |
| Total duration of depression (months), mean (SD) | 36.7 (64.7) | 24.6 (29.7) | 23.6 (33.3) | 15.3 (15.1) | 0 |
| Use of psychotropic treatment (%) | 93.9 | 83.3 | 90.1 | 48.4 | 0 |
| MINI 5.0.0 (%) | | | | | |
| • MDE, current episode | 100 | 0 | 100 | 0 | 0 |
| • MDE, lifetime | 100 | 100 | 100 | 100 | 0 |
| • Suicidal risk, last month | 75.8 | 0 | 75.8 | 0 | 0 |
| • (Hypo)-mania, lifetime | 100 | 100 | 0 | 0 | 0 |
| • Panic disorder, lifetime | 15.2 | 13.3 | 12.1 | 12.9 | 2 |
| • Agoraphobia, current episode | 36.4 | 3.3 | 12.1 | 12.9 | 8.2 |
| • Social phobia, current | 24.2 | 10 | 30.3 | 9.7 | 4.1 |
| • GAD, last 6 months | 48.5 | 20 | 60.6 | 6.5 | 4.1 |
| • OCD, last month | 3 | 0 | 6.1 | 0 | 0 |
| • PTSD, last month | 3 | 0 | 9.1 | 0 | 0 |
| • Alcohol abuse, last 12 months | 0 | 0 | 6.1 | 0 | 0 |
| • Cannabis abuse, last 12 months | 0 | 3.3 | 0 | 0 | 0 |
| • Psychotic disorder, lifetime | 0 | 0 | 0 | 0 | 0 |
| • Eating disorders, last 3 months | 0 | 0 | 0 | 3.2 | 0 |
| MADRS, mean (SD) | 41.3 (8.3) [B] | 2.0 (2.0) [A] | 39.3 (8.4) [B] | 2.2 (2.2) [A] | 1.6 (2.9) [A] |
| YMRS, mean (SD) | 0.6 (1.6) [A] | 0.8 (1.4) [A] | 0.5 (0.9) [A] | 0.1 (0.4) [A] | 0.1 (0.5) [A] |
| Physical anhedonia, mean (SD) | 24.0 (8.7) [B] | 16.4 (9.5) [A] | 25.0 (9.0) [B] | 17.0 (9.0) [A] | 14.0 (7.1) [A] |
| Social anhedonia, mean (SD) | 18.8 (6.2) [C] | 14.4 (7.8) [B] | 19.6 (6.0) [C] | 13.0 (5.6) [AB] | 9.9 (5.7) [A] |
| STAI-trait, mean (SD) | 60.7 (9.3) [B] | 44.3 (10.6) [A] | 55.4 (11.1) [B] | 44.3 (8.6) [A] | 39.8 (9.5) [A] |
| STAI-state, mean (SD) | 61.1 (14.3) [B] | 31.3 (9.0) [A] | 57.8 (15.6) [B] | 31.7 (9.3) [A] | 31.4 (9.0) [A] |

DB: depressed bipolar; EB: euthymic bipolar; DU: depressed unipolar; EU: euthymic unipolar; HC: healthy controls. MINI 5.0.0: Mini-International Neuropsychiatric Interview version; MADRS: Montgomery Åsberg Depression Rating Scale; STAI: State and Trait Anxiety Inventory; YMRS: Young Mania Rating Scale. For each clinical and psychometric parameter, if means share the same letter, they are not significantly different at the 5% level of significance (Tukey test).

## Olfactory identification

For the odors' identification performances on the Sniffin' Sticks–Screening 12 test, a significant group effect was observed ($F_{(4,171)} = 4.4$, p = 0.002, $\eta^2 = 0.09$). The Tukey post hoc test indicated that DU patients identify significantly less odors than EU patients and controls. Identification scores of DB and EB groups were not significantly different from other groups (Fig 2).

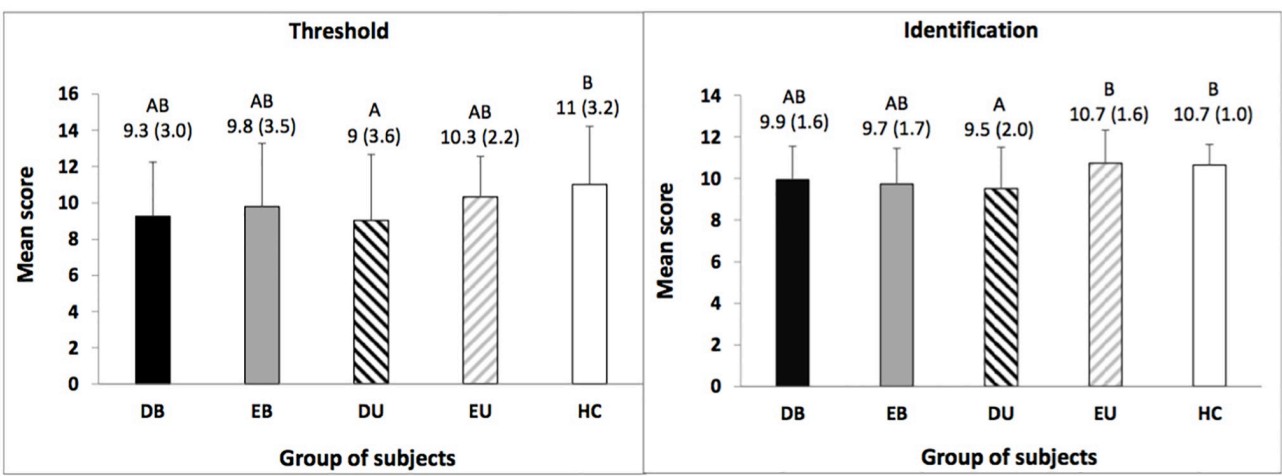

**Fig 2. Mean threshold and identification scores in Depressed Bipolar (DB), Euthymic Bipolar (EB), Depressed Unipolar (DU), and Euthymic Unipolar (EU) patients compared to Healthy Controls (HC).** For each parameter, the means with the same letters are not significantly different at the 5% level of significance (Tukey test). Values in parentheses and error bars indicate standard deviation.

There was no significant difference between groups in the identification of negative odors ($\chi^2 = 5.6$; df = 4; p = 0.2). As for positive odors, DB, EB and DU patients identified significantly less positive odors than controls, and DU patients identified significantly less positive odors than EU patients ($\chi^2 = 33$; df = 4; p<0.001), (Fig 3). Statistics of the post hoc tests (p-values and effect sizes) concerning odors identification are reported in the S1–S13 Tables.

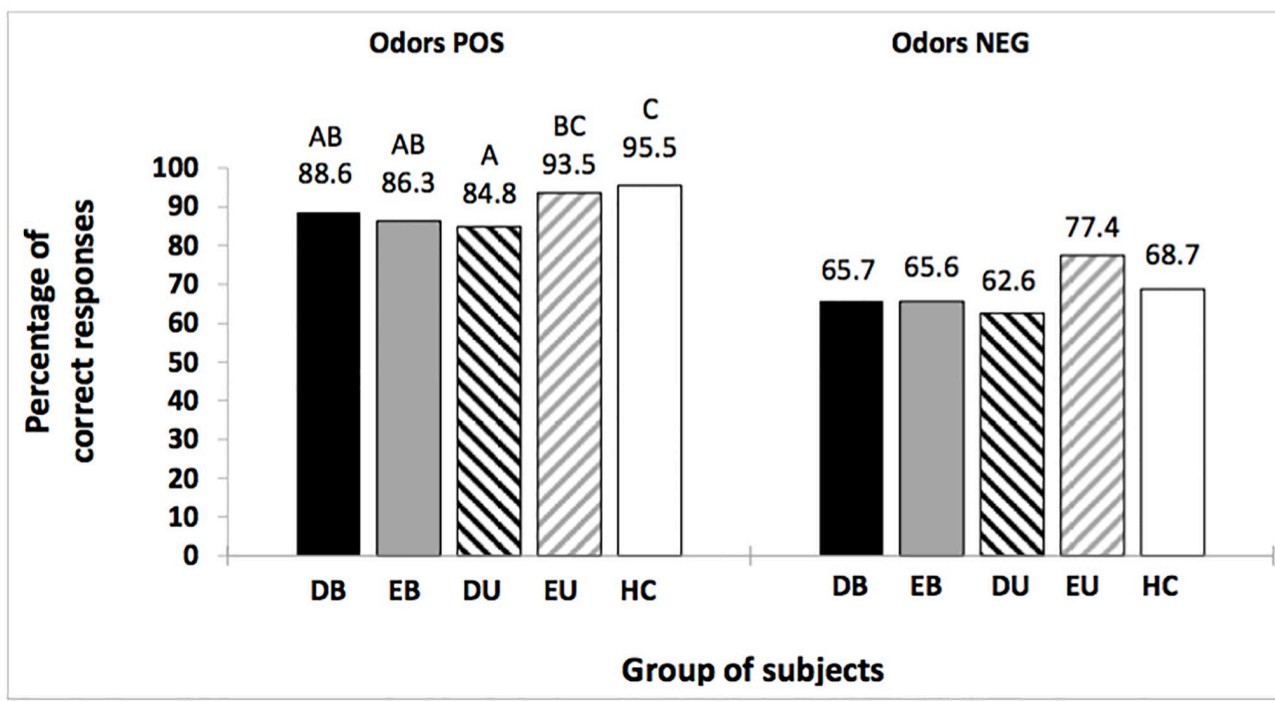

**Fig 3. Between-group comparisons of the number of correct identification responses for Positive (POS) and Negative (NEG) odors.** DB: depressed bipolar; EB: euthymic bipolar; DU: depressed unipolar; EU: euthymic unipolar; HC: healthy controls. For positive odors, values with the same letters are not significantly different at the 5% level of significance (Marascuilo procedure).

**Table 2. Mean scores (standard deviation) for the olfactory judgments (all odors) evaluated by Depressed Bipolar (DB), Euthymic Bipolar (EB), Depressed Unipolar (DU), Euthymic Unipolar (EU) patients compared to Healthy Controls (HC).**

|  | DB (n = 33) | EB (n = 30) | DU (n = 33) | EU (n = 31) | HC (n = 49) |
|---|---|---|---|---|---|
| **Olfactory judgement** |  |  |  |  |  |
| Pleasantness, mean (SD) | 4.4 (3.4) | 5.8 (3.6) | 5.4 (3.2) | 6.1 (3.1) | 6.3 (2.9) |
| Familiarity, mean (SD) | 5.9 (3.6) | 6.8 (3.3) | 6.4 (3.3) | 7.9 (2.5) | 7.6 (2.5) |
| Intensity, mean (SD) | 6.7 (3.0) | 7.4 (2.6) | 6.9 (2.6) | 7.2 (3.1) | 6.9 (2.0) |
| Emotion, mean (SD) | 5.2 (3.6) | 5.8 (3.6) | 5.6 (3.3) | 6.0 (2.0) | 6.0 (2.6) |

### Pleasantness, familiarity, intensity, and emotional rating of odors

The two way analysis of variance with interaction, indicated a significant effect of stimulus for all odor's characteristic except intensity (Pleasantness: $F_{(1,2102)} = 537$, p<0.001, $\eta^2 = 0.19$; Familiarity: $F_{(1,2102)} = 244$, p<0.001, $\eta^2 = 0.1$; Intensity: $F_{(1,2102)} = 0.5$, p = 0.46, $\eta^2 < 0.001$; Emotion: $F_{(1,2102)} = 479$, p<0.001, $\eta^2 = 0.18$). A significant group effect was found for all olfactory parameters (Pleasantness: $F_{(4,2102)} = 14.6$, p<0.001, $\eta^2 = 0.02$; Familiarity: $F_{(4,2102)} = 30$, p<0.001, $\eta^2 = 0.05$; Intensity: $F_{(4,2102)} = 5$, p<0.001, $\eta^2 = 0.01$; Emotion: $F_{(4,2102)} = 5$, p = 0.001, $\eta^2 = 0.008$). Table 2 shows the mean ratings of different groups for the pleasant, familiar, intense and emotional aspect of the 12 odors of the Sniffin' Sticks–Screening 12 identification test.

With regard to the "group x stimulus" interaction, the results showed a difference between the groups according to the stimulus for pleasantness only (Pleasantness: $F_{(4,2102)} = 5.9$, p<0.001, $\eta^2 = 0.004$; Familiarity: $F_{(4,2102)} = 2.3$, p = 0.06, $\eta^2 = 0.004$; Intensity: $F_{(4,2102)} = 1$, p = 0.3, $\eta^2 = 0.002$; Emotion: $F_{(4,2102)} = 0.8$, p = 0.52, $\eta^2 = 0.001$). As for the pleasantness score reflecting hedonic rating of odors, significant difference was found between groups concerning the rating of pleasant positive (POS) odors only (Fig 4). DB patients rated the odors significantly less pleasant the 4 other groups (DU, EB, EU and HC). DU patients had also significantly lower hedonic ratings than Controls and EU patients. Statistics of the post hoc tests (p-values and effect sizes) concerning hedonic perception of odors are reported in the S1–S13 Tables.

### Correlation between hedonic rating and clinical variables

A significant negative correlation was demonstrated between social anhedonia score and the hedonic rating of pleasant odors for EB patients (r = -0.37, p = 0.04).

As for EU patients, significant negative correlation coefficients were found between the hedonic rating of pleasant odors and STAI-state score (r = -0.425, p = 0.017), physical anhedonia (r = -0.43, p = 0.015) and social anhedonia (r = -0.38, p = 0.034) scores respectively (Fig 5).

### Receiver operating characteristic (ROC) curve analysis

In Table 3, the results of the ROC analysis are shown for all olfactory tests. Concerning olfactory threshold, the value of the area under the ROC curve (AUC), which indicates how well the threshold score is able to discriminate between patients and healthy controls, was highest for DB and DU (0.66) groups and lowest for EU group (0.59). The pairwise comparisons with Student tests indicated no significant difference between the AUC of the four groups. The results of the z-tests demonstrated that for DB and DU groups, the AUC were significantly different from 0.5. It was not the case for EB and EU groups.

For each variable, the AUC (Area Under the Curve) with the same letters are not significantly different at the 5% level of significance (Student test with Bonferroni correction). Each

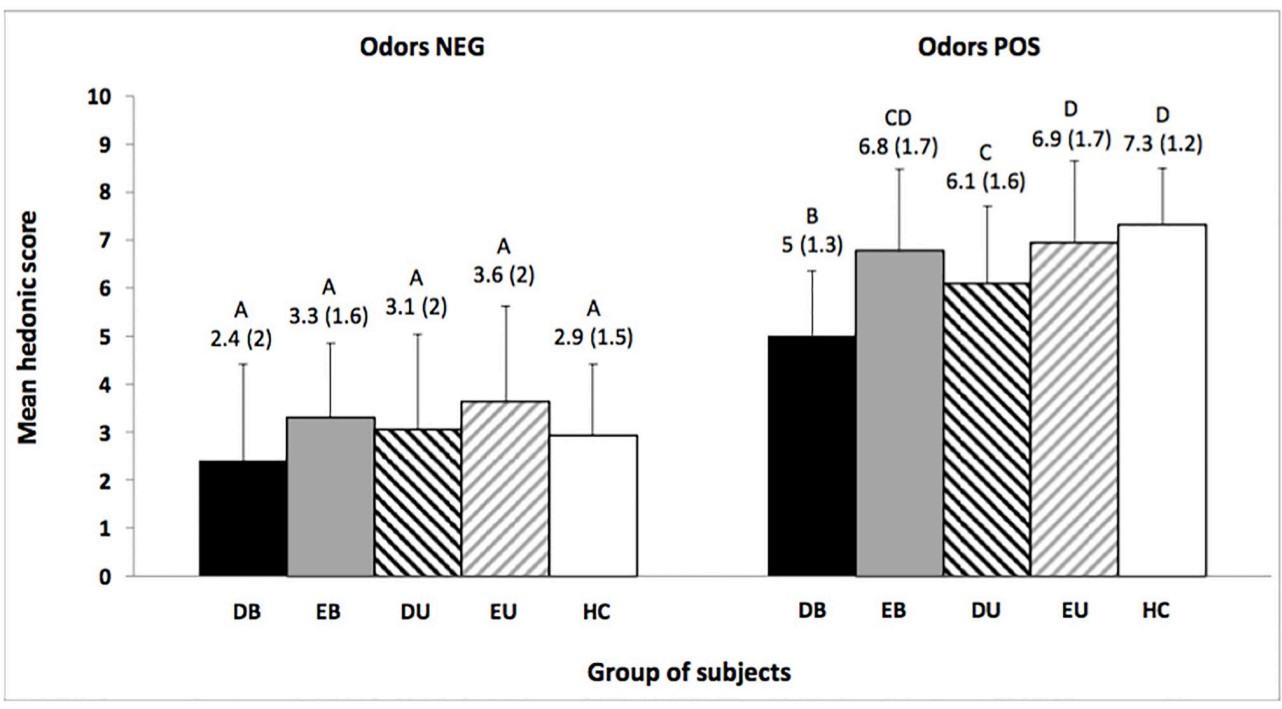

**Fig 4. Mean hedonic scores for the positive (NEG) and the negative (POS) odors.** DB: depressed bipolar; EB: euthymic bipolar; DU: depressed unipolar; EU: euthymic unipolar; HC: healthy controls. Means with the same letters are not significantly different at the 5% level of significance (Tukey test). Values in parentheses and error bars indicate standard deviation.

AUC was compare to 0.5 using z-test. CI: confidence interval, ROC: Receiver operating characteristic, DB: Depressed Bipolar, DU: Depressed Unipolar, EB: Euthymic Bipolar, EU: Euthymic Unipolar.

Regarding odors' identification test, the AUC of the EU group was significantly lower compared to DB, DU and EB groups. The AUC for DU and EB groups only were significantly different from 0.5.

For pleasantness rating of positive odors (POS), the AUC was highest for DB group (0.70) and lowest for EB group (0.51). Only the AUC for both symptomatic groups (DB and DU) were significantly different from 0.5. Concerning the pleasantness rating of negative odors (NEG) the DB group has the highest AUC (0.58) and for this group only, the AUC was significantly different from 0.5.

As for familiarity, the AUC of DB group was highest (0.62). The results of the z-tests revealed that for all groups the AUC were significantly different from 0.5; but for EU group the AUC was lowest than 0.5 (0.46). Regarding intensity, the AUC for DB and DU groups were not significantly different from 0.5. The AUC for EB and EU groups were significantly different from 0.5, but lower than 0.5. At last, concerning emotional rating, the AUC of DB group (0.56) was significantly higher compared to other three groups and for this group only the AUC was significantly different from 0.5.

## Discussion

In the present study, we have assessed olfactory performance of four groups of patients in both unipolar and bipolar depression, in symptomatic and euthymic states. We have compared the performances of these groups between them and with healthy controls. The olfactory functions

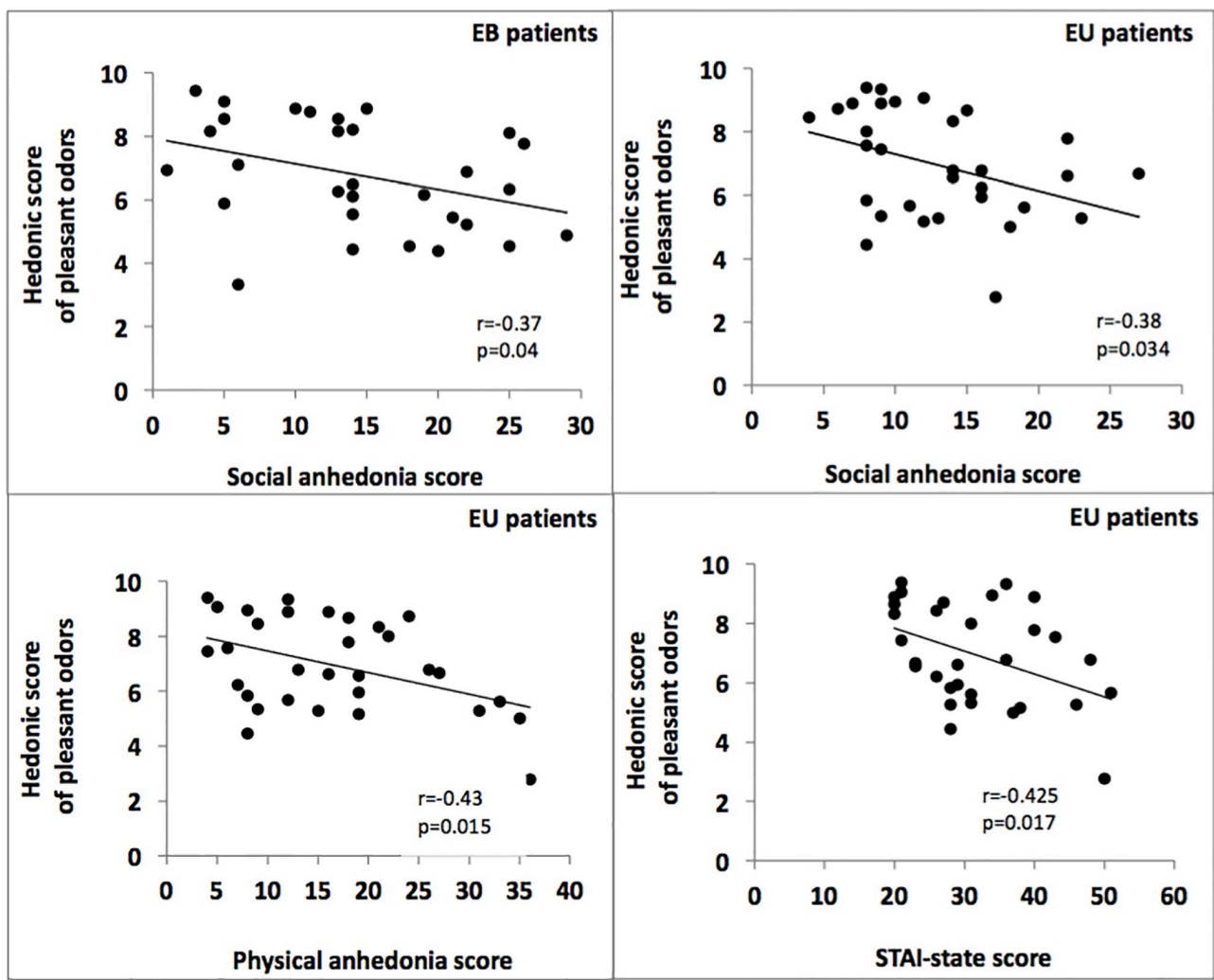

**Fig 5. Scatterplot for the correlation data.** Correlation between hedonic rating and clinical variables (Pearson coefficient).

evaluated in this study were the olfactory threshold, olfactory identification and participants' ratings of pleasantness (hedonic rating), intensity, familiarity and emotion for different odors. Thus, the study aims at giving results concerning state and trait olfactory alterations associated with unipolar and bipolar depression.

## Olfactory threshold

The olfactory threshold measured in our study was higher in DU patients compared to controls, meaning that unipolar depressed patients were less sensitive to odors than healthy individuals. The results were not significantly different among groups of patients. However, we have observed a tendency of progressive increase in odor sensitivity between patients in depressed and euthymic states. These results are in accordance with literature showing a decrease in olfactory sensitivity in depressive states [14–18]. This relationship between depressive symptoms and odor sensitivity can be attributed to close connections between the olfactory bulb, the olfactory sulcus and the amygdala [17, 44–46].

**Table 3. Results of ROC analysis for all patients' groups (DB, EB, DU and EU) and olfactory tests.**

| Olfactory test | AUC | Standard deviation | 95% CI | p-value (z-test) |
|---|---|---|---|---|
| **Olfactory threshold** | | | | |
| DB | 0.66 [A] | 0.07 | 0.54–0.79 | 0.013 |
| DU | 0.66 [A] | 0.07 | 0.53–0.80 | 0.020 |
| EB | 0.61 [A] | 0.07 | 0.46–0.75 | 0.150 |
| EU | 0.59 [A] | 0.07 | 0.45–0.73 | 0.193 |
| **Identification** | | | | |
| DB | 0.62 [B] | 0.07 | 0.49–0.76 | 0.068 |
| DU | 0.67 [B] | 0.06 | 0.54–0.79 | 0.009 |
| EB | 0.64 [B] | 0.07 | 0.50–0.78 | 0.046 |
| EU | 0.42 [A] | 0.07 | 0.29–0.56 | 0.261 |
| **Pleasantness (POS)** | | | | |
| DB | 0.70 [D] | 0.02 | 0.65–0.75 | < 0.0001 |
| DU | 0.61 [C] | 0.02 | 0.57–0.66 | < 0.0001 |
| EB | 0.51 [A] | 0.03 | 0.45–0.56 | 0.800 |
| EU | 0.53 [A] | 0.03 | 0.48–0.58 | 0.269 |
| **Pleasantness (NEG)** | | | | |
| DB | 0.58 [C] | 0.04 | 0.50–0.66 | 0.046 |
| DU | 0.46 [AB] | 0.04 | 0.38–0.54 | 0.359 |
| EB | 0.51 [B] | 0.04 | 0.43–0.60 | 0.747 |
| EU | 0.44 [A] | 0.04 | 0.36–0.52 | 0.128 |
| **Familiarity** | | | | |
| DB | 0.62 [D] | 0.02 | 0.58–0.66 | < 0.0001 |
| DU | 0.59 [C] | 0.02 | 0.55–0.63 | < 0.0001 |
| EB | 0.54 [B] | 0.02 | 0.50–0.59 | 0.047 |
| EU | 0.46 [A] | 0.02 | 0.42–0.50 | 0.041 |
| **Intensity** | | | | |
| DB | 0.49 [C] | 0.02 | 0.45–0.54 | 0.775 |
| DU | 0.48 [C] | 0.02 | 0.44–0.52 | 0.378 |
| EB | 0.42 [A] | 0.02 | 0.38–0.47 | 0.001 |
| EU | 0.45 [B] | 0.02 | 0.41–0.49 | 0.015 |
| **Emotional rating** | | | | |
| DB | 0.56 [C] | 0.02 | 0.51–0.60 | 0.008 |
| DU | 0.53 [B] | 0.02 | 0.49–0.57 | 0.134 |
| EB | 0.49 [A] | 0.02 | 0.45–0.54 | 0.830 |
| EU | 0.50 [A] | 0.02 | 0.46–0.54 | 0.971 |

The value of the area under the ROC curve (AUC), which is a measure of how well the threshold score is able to discriminate between patients and healthy controls, is the same (0.66) for both symptomatic groups (DB and DU) and no significant difference was found between the AUC of the four groups. However, the AUC values of two groups of euthymic patients are not significant different from 0.5, showing that there is no discrimination of these two groups of patients with controls. Obviously, our results demonstrate that the reduction in odor sensitivity is specific for unipolar depressive states, since the results show that this group of patients recover their odor sensitivity after symptomatic remission. This observation concerning unipolar depression was already reported by other studies [14]. Moreover, when we compared the threshold scores of our four groups with the normative value in the literature (healthy individuals between 31–40 years, threshold score = 8.93±2.87) [47], no significant

difference was found for DB (t = 0.6; df = 32; p = 0.53), for DU (t = 0.14; df = 32; p = 0.89) and for EB (t = 1.3; df = 29; p = 0.19) groups. However, the threshold score of the EU group (t = 3.5; df = 30; p = 0.002) and healthy controls (t = 4.6; df = 48; p<0.001) were significantly higher compared to the normative value. This lag of the results especially concerning the controls group could be due to the differences of the experimental conditions and/or the difference of the cultures. This last point is discussed later in the manuscript. It must be also noticed, that globally, the discriminatory power of our ROC analysis was low.

## Olfactory identification

Our results show that odor identification capacity was significantly lower in DU patients compared to controls. Moreover, the accuracy to which the identification test predicts whether an individual belongs to the healthy controls or to DU groups is relatively high (0.67) and significantly different from 0.5. When considering pleasant odors, these differences are more pronounced showing that DU, DB and EB patients have a reduction in their odor identification capacity. Therefore, identification seems to be altered in both unipolar and bipolar depressive states, but only unipolar patients would recover a regular identification level in remission phase, while bipolar patients would keep their deficits even after symptomatic remission. When comparing the identification scores of our four groups with the normative value in the literature (healthy individuals between 31–40 years, identification score = 10.6±1.8) [43], no significant difference was found for EU (t = 0.5; df = 30; p = 0.62) and HC (t = 0.4; df = 48; p = 0.71) groups. But for our DB (t = -2.3; df = 32; p = 0.025), DU (t = -3.1; df = 32; p = 0.004) and EB (t = -2.8; df = 29; p = 0.01) groups, the identification score were significantly lower compared to the normative value. These observations strengthen even more our results discussed above. Only few studies in literature show that odor identification is altered in depression [22]. In many studies, significant alteration in odor identification in depression is not found [15, 20, 21, 24, 48–51].

Other studies show that odor identification is altered in Alzheimer disease [13]. Impairment in cognitive functions observed in both Alzheimer Disease and depressive disorders may explain that odor identification deficit is seen in both disorders. Differences between studies concerning odor identification in depression can also be explained by population and cultural differences between samples. Smell ability can be affected by habituation, stimulation and may differ between different cultures. Studies show that olfactory perception varies between societies even if they share the same language. These variations depend on the cultural specific knowledge of each society [52]. Furthermore, affective responses to odors vary between countries. Ferdenzi et al (2013) report that subjects from Singapore found that odors were less familiar, less intense and less pleasant than European subjects [53]. Our study is the first one to be conducted on a Lebanese Mediterranean population, while all others included either American or European (France, Germany, Poland) subjects. This difference may explain the association between odor identification and depression observed in our study and not in others. However, this association is still unclear and more investigations of the relationship between odor identification and depression are needed.

Odor identification is dependent on several cognitive factors including semantic memory, denomination capacities and understanding of the instructions [13]. The orbitofrontal cortex (OFC) is involved in odor identification, in the judgment of the hedonic value of odors [54], and in cognitive impairments associated with depression [13, 55]. Several parts of the OFC may be implicated in both olfactory and depressive processes. The medial OFC is activated by pleasant odors, while the posterior mid-orbitofrontal cortex is activated by unpleasant components of odors [56]. Theories also suggest that in depression the lateral OFC is activated and

 

the medial OFC has a decreased activity, both participating to the cognitive symptoms of depression [57, 58]. Other brain areas including the hippocampus and the amygdala are also involved in odor identification [59]. Dysfunctions in these areas and associated cognitive impairments may explain the identification deficits observed in our study in depressed states. The hippocampus is activated in odor memorization [60]. The volume of this structure is also decreased in depression [61]. As for the amygdala, it plays a role in the memorization of the emotional aspect of odors [45]. Several studies have also showed an abnormal activity of the amygdala in depression. Their results are however contradictory since some show an increased volume in depression [62], while others are in favor of a decreased volume of this brain structure [63].

Overall, odor identification depends on several cognitive features and different brain areas. Olfactory function deficits and depressive symptoms may be associated to the same brain regions. However, this overlap in anatomical brain regions does not necessarily imply common pathophysiology. The links and connections between these different functions and areas are still uncertain, thus needing further research to fully understand odor identification capacity. In order to understand the association between these overlapping functions, fMRI studies may be needed. Functional imagery assessment was not a part of our study design, thus reducing the extent of our results' interpretation.

Odor identification deficits found in our study are more pronounced regarding pleasant odors. Patients with bipolar disorder show deficits in identifying pleasant odors during depression and after symptomatic remission. These results are confirmed by Lahera et al (2016), showing deficit in olfactory identification in euthymic bipolar patients. The persistence of these identification deficits in euthymic bipolar patients may be a possible indicator of the persistence of cognitive alterations and deficits in emotional perception in bipolar disorders after remission [27]. Moreover, our results of the ROC analysis revealed that the accuracy of the identification test prediction concerning the distinction between healthy controls and DB patients was not more powerful than just a random rule.

## Pleasantness of odors and anhedonia

Studies show that patients with depression exhibit altered hedonic rating of odors compared to controls [14, 15, 23]. Our results show that depressed patients have a lower hedonic rating of pleasant odors (POS) compared to euthymic patients and to healthy controls. Moreover, hedonic rating is lower in bipolar depressed patients compared to unipolar ones, and these deficits improve after remission. Therefore, the hedonic rating of pleasant odors can distinguish bipolar depressions from unipolar depressions during periods of decompensation and during the phases of remission. Our results of the ROC analysis support these observations. Indeed, the highest AUC value was obtained for the hedonic POS rating of DB group (0.70) and for EB group the distinguishing between patients and controls could not be performed (AUC = 0.51).

This rating of pleasantness is a feature of the orbitofrontal representation modulated by affective states. It also depends on the integrative function of the prefrontal cortex [18, 64]. Dysfunctions in these brain regions observed in depression can explain these deficits in hedonic ratings [10]. Therefore, hedonic rating of pleasant odors can constitute a possible indicator for depressive states, but also a potential differentiator between unipolar and bipolar depression. A complementary ROC analyses based on DU and DB groups division revealed a relatively low accuracy (AUC = 0.59) of distinction of these two groups concerning the rating of the pleasant odors, but it is significantly higher than 0.5 (z = 3.9; p<0.001).

 

Clinical anhedonia is a major criterion in the diagnosis of depressive episodes [1]. A relationship between clinical and sensory anhedonia has been established. Berlin et al. (1998) show that anhedonia can be expressed on a gustatory level [65], while Atanasova et al. (2010) demonstrate the presence in depressed subjects of olfactory anhedonia of pleasant odors on qualitative and quantitative levels [23]. Our study is replicating such results by showing marked olfactory anhedonia of positive odors in depressed unipolar and bipolar subjects. The relationship between olfactory and clinical anhedonia is also expressed in our study through the correlations between these variables. Negative correlations between hedonic rating of positive odors and the Chapman anhedonia questionnaire scores are found in EU and EB groups. This shows that olfactory anhedonia is a direct reflection of clinical anhedonia and of the emotional state of the subject. This association between olfactory and clinical anhedonia is detected in remitted patients in euthymic states. Our results show a significant difference between groups in olfactory identification and hedonic rating of pleasant (positive) odors only. This result is due to olfactory anhedonia for pleasant stimuli seen in depressed subjects [23]. As for unpleasant (negative) odors, the presence in our test of only 3 odors (compared to 9 positive odors), may have prevented the appearance of significant difference between groups. A future use of olfactory tests with more unpleasant odors may show more significant differences between groups.

## Neuroimaging in depression and olfaction

Studies have shown that depressed states are associated to abnormal activations in different brain regions including the orbitofrontal cortex, the prefrontal cortex, the amygdala and the anterior cingulate. Considering that these regions are also involved in olfactory perception, their dysfunctions in bipolar and depressive disorder would have implications on olfactory function [10, 18]. Structural abnormalities of olfactory structures are also observed in depressed subjects. Rottstädt et al. (2018) observed a reduction in olfactory bulb volume in depressed patients. The volume of the olfactory bulb in depression was correlated to the volume of the insula, superior temporal cortex and amygdala [66]. These results are confirmed by Negoias et al. (2016) that observed a correlation between olfactory bulb volume and depression severity [44]. This structure may indeed constitute a biological vulnerability factor for the occurrence and maintenance of depression [44, 67]. Structural abnormalities are also seen in other brain structures. Depressed subjects (in symptomatic and remission phases) have shallower olfactory sulci compared to controls suggesting that abnormal olfactory sulcus morphology may be a trait-related marker of vulnerability to depression [46]. Studies using functional brain MRI showed that subjects with olfactory impairment have reduced right hippocampal brain responses to emotional stimuli [68]. Takahashi et al. (2010) also showed that current and remitted depressed subjects have reduced left anterior insular cortex volume, a possible trait marker of depression [69]. This region plays a major role in emotional regulation and olfactory discrimination [69, 70]. All these abnormalities of olfactory structures in depression can explain the deficits in olfactory sensitivity, identification and hedonic appraisal seen in our population.

In summary, this study shows that depressed patients express deficits in the hedonic aspect of their olfactory perception, as well as alterations in olfactory identification and sensitivity. Our study has also showed that major olfactory differences are found between unipolar and bipolar depressed subjects. Olfactory threshold and global identification of smells were affected only in unipolar patients and not in bipolar ones. However, as mentioned earlier, we observed a tendency of progressive increase in odor sensitivity between patients in depressed and euthymic states. But when assessing the identification of Positive (Pleasant Odors), both types of

depression were affected, but only unipolar subjects recovered their deficit after remission. These differences in olfactory measures between unipolar and bipolar depression compared to healthy controls can be related to several factors. These results may be indicators of the patho-physiological differences between unipolar and bipolar depression. Reduction in olfactory bulb volume observed in depression can cause deficit in olfactory sensitivity and identification capacity [17, 44]. However, studies have showed that mood stabilizers (Lithium and valproate) widely used in patients with bipolar disorder may have a neuroprotective effect by preventing dopamine depletion in the olfactory bulb and striatum [71]. This may explain why some olfactory deficits are observed in unipolar depressed subjects and not bipolar ones. Deficit in olfactory identification was noted in both unipolar and bipolar depressed subjects regarding only pleasant smells. Olfactory identification involved cognitive processes [13] that may be altered in depression. Studies show that depressed patients exhibit abnormal reactivity of the amygdala and a decreased response to positive stimuli thus explaining deficit in the olfactory identification of pleasant smells [72, 73].

This is the first study that shows differences in olfactory function between bipolar and unipolar depression. Besides being potential indicators of depression, some olfactory alterations may help differentiating between unipolar and bipolar depression. In our study, the hedonic rating of pleasant odors was found to be a possible indicator of depressive state and a potential factor differentiating between unipolar and bipolar phases. The results of this study add to the available literature that shows that olfaction may be an objective tool to evaluate depressive disorders in symptomatic and euthymic phases. It also shows evidence that sensory assessments can help differentiate between bipolar and unipolar depression, adding therefore accuracy to the diagnostic and treatment processes.

## Limitations

Some limitations of the present study merit discussion. First, this is a cross-sectional study comparing the olfactory function of different groups of patients in depressive and euthymic phases. A more accurate evaluation would have been a prospective comparison of the same patients in the depressive phase and after remission. To reduce this bias, we have matched all our participants' groups on age, sex and smoking status. Second, the effect of treatment on olfaction was not studied. All depressed patients and most euthymic patients were taking psychotropic medications. These medications may have a possible effect on olfaction (although taking a medication with direct effect on olfactory perception was an exclusion criterion) [10] and may constitute a bias in our study. Third, clinical evaluation of patients included taking history of past mood episodes, duration of episodes, number and duration of hospital admissions. This information may be subject to patients' recall bias that should be acknowledged. Fourth, as suggested previously in this part, odors' identification, odors' familiarity and pleasantness may be subject to cultural differences depending on the extent of an odor's use in specific populations. This study was conducted in Lebanon, and the results observed in Lebanese patients may differ from those of other clinical populations. In this study we used Arabic validated versions of the MINI, STAI and MADRS scales. However, we couldn't find Arabic validated versions of the YMRS and Chapman Physical and Social Anhedonia Questionnaire. For the YMRS, we used an Arabic translation of this scale. However, this scale was used to rule out manic/hypomanic episodes that were already ruled out by the MINI. As for the Chapman Anhedonia Questionnaire, we did an Arabic translation and a back translation of this questionnaire. The results obtained in this study were gathered for the validation of the Arabic version of this questionnaire that is still in process. This study evaluated subjects with bipolar disorders. However, we didn't evaluate subjects in manic or hypomanic states. Studies have

showed that bipolar subjects exhibit a characteristic magnification and persistence of their smell abilities in manic/hypomanic phases [28, 29]. An evaluation of patients during these phases would have brought new data to our study and improved our understanding of olfactory changes in bipolar disorders. Subjects in our study had also different number of mood episodes and hospital admissions. Significant differences were seen between groups, and these variables may constitute a bias in the interpretation of olfactory differences. A future study including subjects with the same number of mood episodes would provide more consistent and accurate data on olfactory function in depressive and euthymic states. Patients with severe cognitive impairment were excluded from the study because of their incapacity to complete the assessment. Therefore, severely depressed patients with cognitive impairment may have been excluded from this study, thus affecting its results. At last, we should mention that our sample was recruited from a clinical population in two hospital settings, and might not result of true randomization of subjects presenting with depressive symptoms in the general population. These biases should be acknowledged before generalizing the results of our study.

## Conclusion

In conclusion, our results have demonstrated the presence of potential olfactory indicators regarding states and traits of unipolar/bipolar depression. These olfactory indicators may also help in differentiating unipolar depression from bipolar one, and may help in determining future olfactory markers of depression. However, these results need to be replicated in future studies in order to specify these possible markers and to determine cut-off variations in olfaction that can be used on an individual level to contribute to the diagnosis of depression.

## Supporting information

**S1 Table. Demographic and clinical characteristics of patients: Depressive episodes.** Two-by-two comparisons between groups using Tukey test. α = 0.05 (DB: depressed bipolar patients. n = 33; EB: euthymic bipolar patients. n = 30; DU: depressed unipolar patients. n = 33; EU: euthymic unipolar patients. n = 31 and HC: healthy controls. n = 49). d: Cohen's effect size.
(DOCX)

**S2 Table. Demographic and clinical characteristics of patients: Hospital admissions.** Two-by-two comparisons between groups using Tukey test. α = 0.05 (DB: depressed bipolar patients. n = 33; EB: euthymic bipolar patients. n = 30; DU: depressed unipolar patients. n = 33; EU: euthymic unipolar patients. n = 31 and HC: healthy controls. n = 49). d: Cohen's effect size.
(DOCX)

**S3 Table. Demographic and clinical characteristics of patients: Montgomery Åsberg Depression Rating Scale (MADRS).** Two-by-two comparisons between groups using Tukey test. α = 0.05 (DB: depressed bipolar patients. n = 33; EB: euthymic bipolar patients. n = 30; DU: depressed unipolar patients. n = 33; EU: euthymic unipolar patients. n = 31 and HC: healthy controls. n = 49). d: Cohen's effect size.
(DOCX)

**S4 Table. Demographic and clinical characteristics of patients: Young Mania Rating Scale (YMRS).** Two-by-two comparisons between groups using Tukey test. α = 0.05 (DB: depressed bipolar patients. n = 33; EB: euthymic bipolar patients. n = 30; DU: depressed unipolar

patients. n = 33; EU: euthymic unipolar patients. n = 31 and HC: healthy controls. n = 49).
d: Cohen's effect size.
(DOCX)

**S5 Table. Demographic and clinical characteristics of patients: Physical Anhedonia.** Two-by-two comparisons between groups using Tukey test. $\alpha$ = 0.05 (DB: depressed bipolar patients. n = 33; EB: euthymic bipolar patients. n = 30; DU: depressed unipolar patients. n = 33; EU: euthymic unipolar patients. n = 31 and HC: healthy controls. n = 49). d: Cohen's effect size.
(DOCX)

**S6 Table. Demographic and clinical characteristics of patients: Social Anhedonia.** Two-by-two comparisons between groups using Tukey test. $\alpha$ = 0.05 (DB: depressed bipolar patients. n = 33; EB: euthymic bipolar patients. n = 30; DU: depressed unipolar patients. n = 33; EU: euthymic unipolar patients. n = 31 and HC: healthy controls. n = 49). d: Cohen's effect size.
(DOCX)

**S7 Table. Demographic and clinical characteristics of patients: STAI—Trait.** Two-by-two comparisons between groups using Tukey test. $\alpha$ = 0.05 (DB: depressed bipolar patients. n = 33; EB: euthymic bipolar patients. n = 30; DU: depressed unipolar patients. n = 33; EU: euthymic unipolar patients. n = 31 and HC: healthy controls. n = 49). d: Cohen's effect size.
(DOCX)

**S8 Table. Demographic and clinical characteristics of patients: STAI—State.** Two-by-two comparisons between groups using Tukey test. $\alpha$ = 0.05 (DB: depressed bipolar patients. n = 33; EB: euthymic bipolar patients. n = 30; DU: depressed unipolar patients. n = 33; EU: euthymic unipolar patients. n = 31 and HC: healthy controls. n = 49). d: Cohen's effect size.
(DOCX)

**S9 Table. Odors' identification.** Two-by-two comparisons between groups using Tukey test. $\alpha$ = 0.05 (DB: depressed bipolar patients. n = 33; EB: euthymic bipolar patients. n = 30; DU: depressed unipolar patients. n = 33; EU: euthymic unipolar patients. n = 31 and HC: healthy controls. n = 49). d: Cohen's effect size.
(DOCX)

**S10 Table. Odor threshold.** Two-by-two comparisons between groups using Tukey test. $\alpha$ = 0.05 (DB: depressed bipolar patients. n = 33; EB: euthymic bipolar patients. n = 30; DU: depressed unipolar patients. n = 33; EU: euthymic unipolar patients. n = 31 and HC: healthy controls. n = 49). d: Cohen's effect size.
(DOCX)

**S11 Table. Hedonic scores for the Positive (POS) and the Negative (NEG) odors.** Two-by-two comparisons between groups: Tukey test. $\alpha$ = 0.05 (DB: depressed bipolar patients. n = 33; EB: euthymic bipolar patients. n = 30; DU: depressed unipolar patients. n = 33; EU: euthymic unipolar patients. n = 31 and HC: healthy controls. n = 49). d: Cohen's effect size.
(DOCX)

**S12 Table. Optimal cutoff values.** Correspond to the maximum sum of sensitivity and specificity, when all cutoff values of the variable are considering for each group of subjects. *: Accuracy is the Proportion Correctly Classified: (number of True Positives + number of True Negative) / (number of True Positives + number of False Positives + number of False Negatives + number of True Negative). CI: confidence interval.
(DOCX)

**S13 Table. Pre-test.** Pretest performed on two groups of subjects with similar demographic characteristics to those of the individuals participating in the main experiment. The demographic and psychometric characteristics of the participants are presented in the table below. The results of the odors' identification test (Sniffin' sticks identification test–Screening 12 Test) of the two groups of subjects (subjects with depressive symptoms: DS and Healthy controls: HC) are also reported.
(DOCX)

**S1 Fig. Receiver Operating Characteristic (ROC) curves and results.** Presentation of the ROC curves for each patients' group (DB: depressed bipolar patients; EB: euthymic bipolar patients; DU: depressed unipolar patients; EU: euthymic unipolar patients and HC: healthy controls) concerning all olfactory tests. A receiver operating characteristic (ROC) curve plots the true positive rate (sensitivity) against the false positive rate (1 –specificity) for all possible cutoff values. a. Olfactory threshold. b. Identification. c. Pleasantness (POS). d. Pleasantness (NEG). e. Familiarity. f. Intensity. g. Emotion.
(TIFF)

## Acknowledgments

The authors thank the administration of the Psychiatric Hospital of the Cross and the Psychiatry unit of Hotel Dieu de France hospital for accepting and facilitating the completion of this study in their respective psychiatric units. The authors thank Drs Elie Khoury, Myriam Zarzour and Yara Chamoun for their help in the assessment of included subjects.

## Author Contributions

**Conceptualization:** François Kazour, Sami Richa, Wissam Elhage, Boriana Atanasova.

**Data curation:** Chantale Abi Char, Alexandre Surget.

**Formal analysis:** Alexandre Surget.

**Investigation:** Chantale Abi Char.

**Methodology:** François Kazour, Sami Richa, Wissam Elhage, Boriana Atanasova.

**Project administration:** Sami Richa, Wissam Elhage, Boriana Atanasova.

**Resources:** Wissam Elhage, Boriana Atanasova.

**Supervision:** Wissam Elhage, Boriana Atanasova.

**Validation:** Alexandre Surget, Wissam Elhage, Boriana Atanasova.

**Writing – original draft:** François Kazour.

**Writing – review & editing:** François Kazour, Sami Richa, Alexandre Surget, Wissam Elhage, Boriana Atanasova.

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
