## [Decision Letter · Decision Letter 0]

6 Jan 2020

PONE-D-19-27567

Olfactory markers for depression: differences between bipolar and unipolar patients

PLOS ONE

Dear Dr. KAZOUR,

Thank you for submitting your manuscript to PLOS ONE. After careful consideration, we feel that it has merit but does not fully meet PLOS ONE’s publication criteria as it currently stands. Therefore, we invite you to submit a revised version of the manuscript that addresses the points raised during the review process.

We would appreciate receiving your revised manuscript by Feb 20 2020 11:59PM. To enhance the reproducibility of your results, we recommend that if applicable you deposit your laboratory protocols in protocols.io, where a protocol can be assigned its own identifier (DOI) such that it can be cited independently in the future. For instructions see: http://journals.plos.org/plosone/s/submission-guidelines#loc-laboratory-protocols

We look forward to receiving your revised manuscript.

Kind regards,

Danilo Arnone

Academic Editor

PLOS ONE

Journal Requirements:

3. Please include your tables as part of your main manuscript and remove the individual files.

Please note that supplementary tables should be uploaded as separate "supporting information" files.

Reviewers' comments:

Reviewer's Responses to Questions

**Comments to the Author**

1. Is the manuscript technically sound, and do the data support the conclusions?

Reviewer #1: Partly

Reviewer #2: Partly

2. Has the statistical analysis been performed appropriately and rigorously? 

Reviewer #1: Yes

Reviewer #2: No

3. Have the authors made all data underlying the findings in their manuscript fully available?

Reviewer #1: Yes

Reviewer #2: Yes

4. Is the manuscript presented in an intelligible fashion and written in standard English?

Reviewer #1: Yes

Reviewer #2: Yes

5. Review Comments to the Author

Reviewer #1: Comments to the Authors

In this study, authors investigated difference in olfactory detection and identification between patients with unipolar and bipolar depression, in symptomatic and euthymic state. It is interesting study to observe the difference character for olfactory perception between group of patietns, however, number of issues must be clarify.

Major comments

1. Introduction is too long, and it seems to missing points.

2. Discussion section may be divided into sections.

3. Each results were interesting, however, it was hard to understand the connection between those results. For example, threshold level in DU was significantly lower than HC, on the one hand, identification of DU was significantly lower than EU and HC. These results can understand DU was clearly different from HC but it was not clear DB and EB. It may be discuss such detail difference in discussion with references of the past researches.

4. Recent neuroimaging studies showed olfactory brain activations and volume changes related to level of olfactory ability in psychiatric disorders. It needs to discuss the results obtained this study and possibility of past results of the neuroimaging studies.

5. Why there was no difference between groups for percentage of correct response of unpleasant odor and hedonic scores? But pleasant odor had.

Reviewer #2: I had the plesure to review this manuscript, which I think is a valuable work.

Olfaction and olfactory neurobiology are one of the least researched topics. The authors decided to tackle an area of a relevant gap in knowledge.

In their introduction, the authors were able to conduct a comprehensive review of the literature. They showed how olfaction research might be relevant to mood disorders and how it could be relevant to clinical practice; partially how their research could have the potential to differentiate bipolar from unipolar depression. I think the premise of this work is original.

In the introduction, the authors were able to describe their primary and secondary objectives clearly though the authors could have stated their primary and secondary hypotheses explicitly.

The study had a sound design aimed to examine the questions their sat-out to answer. The choice of groups (DU,DU, EB,EU, and control) was well suited to investigate the difference in olfactory functions between:

those who have mood disorders and those who do not suffer from any mental disorders

between types of bipolar and unipolar mood disorders

those who are in relapse and those who are in remission.

I could not see any rationale for the number of participants, in terms of sample size calculation or power calculations.

The authors were mindful of the potential confounders, such as the effect of smoking and other drugs on olfaction. That reflected on the exclusion criteria.

With regards to the clinical assessment, the authors used the following tools:

1. The Mini International Neuropsychiatric Interview (MINI 5.0.0)

2. Montgomery-Åsberg Depression Rating Scale (MADRS)

3. Young Mania Rating Scale (YMRS)

4. State-Trait Anxiety Inventory (STAI)

5. Chapman physical and social anhedonia questionnaire

The authors did not report the validity of the use of those tests in the Lebanese culture, especially if participants are not native English speakers or if they speak English at all. The authors need to report if the versions of the 5 tools used in the clinical assessment were conducted in English or Arabic. If the authors used Arabic translations, then they need to cite the authors of the Arabic translation of those tools.

They also used The Sniffin' sticks threshold test and Sniffin' sticks identification test – Screening 12 Test for assessing to assess olfactory threshold, olfactory identification capacity and their rating of odours hedonic aspect, familiarity, intensity, and emotional impact. The participant cultural background heavily influences those tests. The authors have already discussion and the limitation of the study.

With regards to the statistical tests used, the authors decided to use one-way analysis of variance (one-way ANOVA) with Tukey posthoc test when comparisons between groups were carried out for quantitative variables. While the use of parametric ANOVA might be suitable for parametric tests, such as age, other variables, such as educational level, total identification score and threshold scores might not conform to the normal distribution probability model. Hence, the use of Kruskal–Wallis one-way analysis of variance might be more suitable. The authors might want to use Dunn's post hoc tests to examine the difference in each pair of groups.

The authors' report of the results was clear in the narrative. They used clear visualisation for the data.

The authors were able to provide a clear summary and clearly explain their findings. Their conclusions followed through the results. That shows the high degree of internal validity of this manuscript. Moreover, those results might be generalisable to the whole population of patients suffering from mood disorders. That adds to the external validity of this work.

In summary, I think this work is original and relevant. It addresses an existing gap in knowledge. Yet in its early stages for it to be clinically applicable. The choice of study groups and the design of the study were sound. The tools used, though might be suitable, might not be validated yet for the non-English speaking population. The authors need to report on the validity of those tools in Lebanese or Arabic population. The authors might want to consider other non-parametric statistical tests for some of the quantitative variables.

I think that is a valuable work that deserves to be published. But the authors need to address the points raised earlier.

6. PLOS authors have the option to publish the peer review history of their article (what does this mean?). If published, this will include your full peer review and any attached files.

Reviewer #1: No

Reviewer #2: No

---

## [Author Response · Author response to Decision Letter 0]

27 Feb 2020

Response to the Reviewers

Reviewer #1:

1. “Introduction is too long, and it seems to missing points.”

Response: Sections of the introduction that are not related directly to the subject were either deleted or reduced following the suggestion of the reviewer. New information was added to Introduction and discussion (clear hypotheses, neuroimaging and anatomical findings) (according to suggestions of reviewers) in order to have a better and comprehensive approach of the subject.

2. “Discussion section may be divided into sections.”

Response: Discussion was divided into sections as suggested by the reviewer

3. “Each results were interesting, however, it was hard to understand the connection between those results. For example, threshold level in DU was significantly lower than HC, on the one hand, identification of DU was significantly lower than EU and HC. These results can understand DU was clearly different from HC but it was not clear DB and EB. It may be discuss such detail difference in discussion with references of the past researches.”

Response: Indeed, as the reviewer noticed major olfactory differences were found between unipolar and bipolar depressed subjects:

-Olfactory threshold and global identification of smells were affected only in unipolar patients and not in bipolar ones.

- Regarding the identification of Positive (Pleasant Odors), both types of depression were affected, but only unipolar subjects would recover their deficit after remission

These olfactory differences between unipolar and bipolar subjects were discussed more clearly in the discussion section as suggested by the reviewer (page 21)

4. “Recent neuroimaging studies showed olfactory brain activations and volume changes related to level of olfactory ability in psychiatric disorders. It needs to discuss the results obtained this study and possibility of past results of the neuroimaging studies.”

Response: As suggested by the reviewer, we used past and recent references to discuss how abnormalities in neuroimaging studies can explain the olfactory deficits seen in depression (p 20)

5. “Why there was no difference between groups for percentage of correct response of unpleasant odor and hedonic scores? But pleasant odor had.”

Response: As suggested by the reviewer, our results show a significant difference between groups in olfactory identification and hedonic rating of pleasant (positive) odors only. This result is due to olfactory anhedonia for pleasant stimuli seen in depressed subjects [Atanasova et al., 2010]. As for unpleasant (negative) odors, the presence in our test of only 3 odors (compared to 9 positive odors), may have prevented the appearance of signifant difference between groups. A future use of olfactory tests with more unpleasant odors may show more significant differences between groups. (This clarification was added to the Discussion page 20).

Reviewer #2: 

1. “In the introduction, the authors were able to describe their primary and secondary objectives clearly though the authors could have stated their primary and secondary hypotheses explicitly.”

Response: As suggested by the reviewer, the primary and secondary hypotheses were stated clearly at the end of the introduction section (page 5).

2. “I could not see any rationale for the number of participants, in terms of sample size calculation or power calculations.”

Response: The sample size calculations were performed based on our preliminary data for olfactory identification and in order to have a number of participants for a statistical power of 0.8. allowing the detection of an effect size eta²=0.06 (i.e. d=0.5) at a significance threshold of 0.05. This information was added to the manuscript on page 8.

3. “With regards to the clinical assessment, the authors used the following tools:

1. The Mini International Neuropsychiatric Interview (MINI 5.0.0)

2. Montgomery-Åsberg Depression Rating Scale (MADRS)

3. Young Mania Rating Scale (YMRS)

4. State-Trait Anxiety Inventory (STAI)

5. Chapman physical and social anhedonia questionnaire

The authors did not report the validity of the use of those tests in the Lebanese culture, especially if participants are not native English speakers or if they speak English at all. The authors need to report if the versions of the 5 tools used in the clinical assessment were conducted in English or Arabic. If the authors used Arabic translations, then they need to cite the authors of the Arabic translation of those tools.”

Response: In this study we used the Arabic validated versions of the MINI (Kadri et al., 2005), the STAI (Hallit et al., 2019) and the MADRS (Hallit et al., 2019). These references were added to the manuscript. As for the YMRS, we used an Arabic translation of this scale since couldn’t find a validated version. However, this scale was used to rule out a manic/hypomanic episodes that were already ruled out by the MINI. As for the Chapman Anhedonia Questionnaire, we did an Arabic translation and a back translation of this questionnaire due to the absence of a validated version. The results obtained in this study were gathered for the validation of the Arabic version of this questionnaire that is still in process. This lack of validation is a limitation of our study and was added to the “Limitations” section of the manuscript.

4. “With regards to the statistical tests used, the authors decided to use one-way analysis of variance (one-way ANOVA) with Tukey posthoc test when comparisons between groups were carried out for quantitative variables. While the use of parametric ANOVA might be suitable for parametric tests, such as age, other variables, such as educational level, total identification score and threshold scores might not conform to the normal distribution probability model. Hence, the use of Kruskal–Wallis one-way analysis of variance might be more suitable. The authors might want to use Dunn's post hoc tests to examine the difference in each pair of groups.”

Response: The normal distribution of the data was not always validated (Kolmogorov-Smirnov test). Despite this, the parametric tests were used. Indeed, based on our sample size (n ≥ 30 per group), conditions for applying the central limit theorem was met, allowing us using one-way ANOVA even when normality may not be ensured for all samples (when central limit theorem applied, sampling distributions follow a normal distribution) (This information was added to the manuscript on page 8).

---

## [Decision Letter · Decision Letter 1]

20 Apr 2020

PONE-D-19-27567R1

Olfactory markers for depression: differences between bipolar and unipolar patients

PLOS ONE

Dear Dr. KAZOUR,

Thank you for submitting your manuscript to PLOS ONE. After careful consideration, we feel that it has merit but does not fully meet PLOS ONE’s publication criteria as it currently stands. Therefore, we invite you to submit a revised version of the manuscript that addresses the points raised during the review process.

We would appreciate receiving your revised manuscript as soon as possible preferably within the next 30 days. To enhance the reproducibility of your results, we recommend that if applicable you deposit your laboratory protocols in protocols.io, where a protocol can be assigned its own identifier (DOI) such that it can be cited independently in the future. For instructions see: http://journals.plos.org/plosone/s/submission-guidelines#loc-laboratory-protocols

We look forward to receiving your revised manuscript.

Kind regards,

Danilo Arnone

Academic Editor

PLOS ONE

Additional Editor Comments (if provided):

It would be great if you were able to address the remaining points made by Reviewer 2.

Reviewers' comments:

Reviewer's Responses to Questions

**Comments to the Author**

Reviewer #1: All comments have been addressed

Reviewer #2: I am pleased to see that the authors have responded adequately to points 1 and 3.

The authors' response to points number 2 and 4 have not, however, addressed the issues raised satisfactorily.

So, for point number 2 on the calculation of the sample size, while authors have reported “statistical power of 0.8. allowing the detection of an effect size eta2=0.06 (i.e. d=0.5) at a significance threshold of 0.05. “, they did not report the parameter they used to calculate the effect size between the different groups. Did the author use the Olfactory threshold or Olfactory identification-related-parameters to calculate the effect-size? More detailed data about the findings of the pilot need to be included.

With regards to the point number 4 concerning the statistical tests used, the authors cited Central Limit Theorem (CLT) and the sample size(n>30) to be sufficient to use parametric tests even if the normality of the distribution of data is not assured. Saying that, CLT requires random sampling from the population of interest. The authors did not report how large was the population they sampled, and if the sampling was done randomly. Hence, without examining the normality of the distribution of the parameters, the authors need to report the details of the sample selection and if this selection was random. The other alternative is to use nonparametric statistics (distribution-free), as mentioned in the previous review.

---

## [Author Response · Author response to Decision Letter 1]

18 May 2020

Reviewer #2: I am pleased to see that the authors have responded adequately to points 1 and 3.

The authors' response to points number 2 and 4 have not, however, addressed the issues raised satisfactorily.

So, for point number 2 on the calculation of the sample size, while authors have reported “statistical power of 0.8. allowing the detection of an effect size eta2=0.06 (i.e. d=0.5) at a significance threshold of 0.05. “, they did not report the parameter they used to calculate the effect size between the different groups. Did the author use the Olfactory threshold or Olfactory identification-related-parameters to calculate the effect-size? More detailed data about the findings of the pilot need to be included.

RESPONSE: We would like to thank the reviewer for their comments. Concerning the comment on the calculation of the sample size, we calculated the statistical power (with XLSTAT) by using means and standard deviations of a previous Olfactory identification task. We performed a pretest on 2 groups of subjects with similar demographic characteristics similar to those of the individuals participating in the main experiment (these results have been added to the supplementary data file: Table S.13). In this pretest we obtained an effect size Cohen's d = 0.9339 (for t-test), which is comparable of a eta² = 0.179 (for ANOVA) (Cohen , 1988). 

For the experiment, we therefore estimated our sample size calculation from a more conservative effect size eta² = 0.06, which corresponds to an effect of intermediate magnitude according to Cohen (1988) and requiring to recruit more participants. Accordingly, the estimation of the sample size for an ANOVA was performed to be able to detect an effect with alpha = 0.05, a power of 0.8 and for an effect size of eta² = 0.06. Hence, the sample size necessary to obtain a power of 0.8 and an effect size of eta² = 0.06 at alpha = 0.05 is 192 participants (if we had kept the less conservative eta² = 0.179 from the pre-test with the same alpha and power, the calculation would have indicated that 60 participants may be enough)In our study, we therefore aimed to recruit more than 192 particpants, and we indeed were able to have 215 participants. However, our criteria made us that we only had 176 subjects have been included in the study.Nevertheless, this number was still much higher than 60. Moreover, with a sample size of 176 we should be able to detect an effect size of 0.065 (with alpha = 0.05 and a power of 0.8, see below), still very sensitive and a more precise effect size than eta² = 0.179.

Parameters - Data input 

Nb of groups : 5

partiel Eta² : 0,065

Parameters - Results

Power 0,8

Alpha 0,05

Effect size (f) 0,26366402

Sample size 176

Power (obtained)0,799

With regards to the point number 4 concerning the statistical tests used, the authors cited Central Limit Theorem (CLT) and the sample size(n>30) to be sufficient to use parametric tests even if the normality of the distribution of data is not assured. Saying that, CLT requires random sampling from the population of interest. The authors did not report how large was the population they sampled, and if the sampling was done randomly. Hence, without examining the normality of the distribution of the parameters, the authors need to report the details of the sample selection and if this selection was random. The other alternative is to use nonparametric statistics (distribution-free), as mentioned in the previous review.

RESPONSE: The reviewer comments on the sampling method of our population and the statistical tests used. In our study, we recruited subjects from two hospital clinical settings: all subjects who presented to inpatients and outpatients clinics with depressive symptoms were approached and then either included or excluded from this study. This method of selection was random even if it did not include a true randomization of all the subjects presenting with depressive symptoms in the general population. This method of sampling is seen in most clinical studies, where strict and true randomization is rarely feasible. Therefore, this leads to a possible selection bias that is a limitation to our study whether we use either parametric or non-parametric tests (Matthews JA, 1981; Scheff SW, 2016). This bias is acknowledged and added to the discussion of the manuscript. In our study and based on the Central Limit Theorem, we considered that the distribution of means approaches normality as sample size is > 30 per group and that sample independence was assumed, and therefore justifying the use of parametric statistics like ANOVA (Kwak & Kim, 2017; ; Scheff SW, 2016). Parametric test like ANOVA gives more powerful results compared to non-parametric Kruskal –Wallis test Finally, our sampling method, despite its limitations, is similar to those used in most clinical studies and allows us to reach results that can only be generalized after taking in consideration the biases mentioned above. We hope that our response provides an adequate answer to the justified concerns of the reviewer regarding the sampling and statistical tests that were used.

• Cohen, J. (1988). Statistical power analysis for the behavioral sciences (2nd ed). Hillsdale, NJ: Erlbaum Associates. DOI:10.4324/9780203771587

• Matthews, J. A. (1981). Further Non-parametric Tests for Independent Samples. In Quantitative and Statistical Approaches to Geography (p. 108‑123). DOI:10.1016/B978-0-08-024295-8.50017-4

• Scheff, S. W. (2016). Fundamental Statistical Principles for the Neurobiologist : A Survival Guide (1 edition). Academic Press. DOI:10.1016/C2015-0-02471-6

• Kwak, S. G., & Kim, J. H. (2017). Central limit theorem : The cornerstone of modern statistics. Korean Journal of Anesthesiology, 70(2), 144‑156. https://doi.org/10.4097/kjae.2017.70.2.144

---

## [Decision Letter · Decision Letter 2]

16 Jul 2020

PONE-D-19-27567R2

Olfactory markers for depression: differences between bipolar and unipolar patients

PLOS ONE

Dear Dr. KAZOUR,

Thank you for submitting your manuscript to PLOS ONE. After careful consideration, we feel that it has merit but does not fully meet PLOS ONE’s publication criteria as it currently stands. Therefore, we invite you to submit a revised version of the manuscript that addresses the points raised during the review process.

We look forward to receiving your revised manuscript.

Kind regards,

Danilo Arnone

Academic Editor

PLOS ONE

Additional Editor Comments (if provided):

Please, review punctuation thorughot the manuscript. Many thanks.

**Comments to the Author**

Reviewer #2: The authors have addressed the two commends raised in the last review. However, table 1 and 3 need reformatting to fit the page.

---

## [Author Response · Author response to Decision Letter 2]

24 Jul 2020

PONE-D-19-27567R2

Olfactory markers for depression: differences between bipolar and unipolar patients 

Dear Editor,

Dear Reviewer,

On behalf of the authors, I would like to thank you for your patience and for the constructive comments you brought to our manuscript.

As requested, the following modifications were made:

Additional Editor Comments (if provided): Please, review punctuation throughout the manuscript. Many thanks.

• Punctuation was reviewed and necessary corrections were made throughout the manuscript

Reviewer #2: The authors have addressed the two commends raised in the last review. However, table 1 and 3 need reformatting to fit the page.

• Tables 1 and 3 were reformatted as requested to fit the page

---

## [Editor Report · Decision Letter 3]

30 Jul 2020

Olfactory markers for depression: differences between bipolar and unipolar patients

PONE-D-19-27567R3

Dear Dr. KAZOUR,

We’re pleased to inform you that your manuscript has been judged scientifically suitable for publication and will be formally accepted for publication once it meets all outstanding technical requirements.

Kind regards,

Danilo Arnone

Academic Editor

PLOS ONE
---

## [Editor Report · Acceptance letter]

4 Aug 2020

PONE-D-19-27567R3 

Olfactory markers for depression: Differences between bipolar and unipolar patients 

Dear Dr. Kazour:

I'm pleased to inform you that your manuscript has been deemed suitable for publication in PLOS ONE. Congratulations! Your manuscript is now with our production department. 

Kind regards, 

on behalf of

Dr. Danilo Arnone 

Academic Editor

PLOS ONE